# Invertebrates of Siberia, a Potential Source of Animal Protein for Innovative Food and Feed Production: Biomass Nutrient Composition Change in the Earthworm *Eisenia fetida* (Savigny, 1826) and the House Cricket *Acheta domesticus* (Linnaeus, 1758)

**DOI:** 10.3390/insects16060632

**Published:** 2025-06-16

**Authors:** Sergei E. Tshernyshev, Andrei S. Babenko, Irina B. Babkina, Ruslan T.-O. Baghirov, Vera P. Modyaeva, Margarita D. Morozova, Ksenia E. Skribtcova, Elena Y. Subbotina, Mikhail V. Shcherbakov, Anastasia V. Simakova

**Affiliations:** Biological Institute, Tomsk State University, Lenina Prospekt, 36, Tomsk 634050, Russia; shib@sibmail.com (A.S.B.); bibsphera@gmail.com (I.B.B.); rbaghirov@yandex.ru (R.T.-O.B.); vera.modyaeva@gmail.com (V.P.M.); science.margarita145@list.ru (M.D.M.); ksu.skriptcova@mail.ru (K.E.S.); orfelia73@mail.ru (E.Y.S.); tephritis@mail.ru (M.V.S.); omikronlab@yandex.ru (A.V.S.)

**Keywords:** terrestrial invertebrates, biomass, nutrient composition

## Abstract

A new method for designing the nutrient composition of invertebrate biomass by enriching the food substrate with precursors is presented. The experiment focused on the content of proteins, minerals, B-group vitamins and liposoluble vitamins (A, E, D and K) in the biomass of the house cricket (*Acheta domesticus* Linnaeus, 1758) and the earthworm (*Eisenia fetida* Savigny, 1826). The results show that various vitamins and minerals increase in the biomass of the model species. The enrichment of the feeding substrate with high doses of precursors may have the opposite effect of reducing some nutrients. The calorific value of crickets is twice that of earthworms and remains unchanged after a double-dose substrate enrichment. It is shown that the nutrient composition of invertebrate biomass can be increased by enriching the feeding substrate with precursors. The most effective increases are observed for all vitamins and several minerals.

## 1. Introduction

The study of the adaptive potential of invertebrates allows scientists to realize analogous mechanisms in engineering, medicine, architecture, agriculture, textiles, the food industry, and many other developed branches of the practical economy [1,2,3,4,5].

One of the current directions of modern agricultural development is the use of terrestrial invertebrates to obtain animal protein products with a nutrient-dense composition (i.e., relatively more nutrients than calories) for feed and food production [6]. The global increase in food demand requires the search for new sources of animal protein in addition to traditional livestock production [7,8,9]. Currently, there is an increasing interest in protein production from terrestrial invertebrates [10,11,12,13,14,15,16,17,18]. The most attention is paid to the so-called deep processing of the material obtained from a farm animal, more precisely the liver, skin, accessible parts of bone tissue, etc. As part of an innovative approach, a new method for the production of snacks from the skin of farm animals, birds, and fish has been presented [19,20,21,22,23,24,25]. In order to compensate for the lack of animal protein and to avoid the problems of environmental pollution, it is necessary to use invertebrate biomass as an alternative to traditional protein sources [26,27,28,29,30,31,32,33,34,35,36], but with reference to the safety of reared edible invertebrates [3,9,11,37,38,39,40].

At present, the trend of rearing terrestrial insects, molluscs, arachnids, worms, and other invertebrates is being developed worldwide [30,41,42,43,44,45,46,47], including in Russia [14,15,16,17]. EC Regulation No 853/2004 defines five species of terrestrial snails as ‘edible snails’ to be used in restaurant gastronomy [48]. One of the most important ways of developing the protein production industry is based on the use of high-quality biomass obtained from terrestrial invertebrates, with its potential to improve the nutrient composition, its high bioavailability, and the bioaccessibility of the main nutrients.

It should be noted that forage and living conditions significantly affect the nutrient composition, especially at the developmental stage, of invertebrate biomass. The implementation of feasible methods of biomass enrichment allows one to obtain the maximum nutrient density product necessary to obtain high-efficiency results in the production of terrestrial invertebrates rearing technology. The directional enrichment of the feeding substrate to obtain the necessary level of certain nutrients in the biomass of model species of invertebrates provide the main theme of the project, which is herein aimed at determining the nutrient composition of designed biomass.

In previous work, the nutrient composition in the biomass of two invertebrate species, the speckled cockroach *Nauphoeta cinerea* (Olivier, 1789) and the giant African land snail *Lissachatina fulica* (Férussac, 1821), was studied and analyzed [49]. The variation in biomass content of two other species, the house cricket *Acheta domesticus* (Linnaeus, 1758) and the earthworm *Eisenia fetida* (Savigny, 1826), is analyzed in this study, aiming to determine the potential and nature of targeted enrichment of invertebrate biomass with required nutrients. The novelty of the research lies in the following aspects: (1) the investigation of changes in the nutrient composition of invertebrates as a result of enrichment of the dietary substrate; (2) the discovery of the uneven accumulation of nutrients in the biomass of different invertebrate species; (3) the identification of nutrient groups that can be most effectively increased; (4) and the determination of the optimal dosage of precursors.

Around 70–80% of livestock productivity is influenced by feeding practices and housing conditions. Among the measures that can be taken to improve livestock productivity, ensuring a balanced diet is of paramount importance. One approach to improving both the productivity and resilience of animals is the use of biologically active substances that act as metabolic agents. The incorporation of bioactive compounds (such as vitamins and minerals) through the biomass of invertebrates increases their bioavailability compared to synthetic forms. This has a direct impact on animal health and productivity [50,51,52]. The results of our research indicate the potential to improve the nutritional value of diets through the incorporation of properly enriched invertebrate biomass, which will have a positive commercial impact by improving animal health, enhancing the palatability of meat, and enriching it with essential nutrients [15,16,17,49,53,54].

The differences in the accumulation of specific nutrients between crickets and earthworms dictate the nature and objectives of enriching the nutritional substrate [53]; for example, earthworms are particularly effective in achieving elevated levels of vitamin C, fat-soluble vitamins (FSV), and minerals, whilst enhancing protein content. In contrast, crickets excel at accumulating B vitamins and essential minerals such as magnesium, phosphorus, potassium, zinc, and iron [45]. The biomass of insects proves to be more advantageous for the accumulation of vitamins and energetically significant minerals. Although insects possess a higher caloric content compared to earthworms, this factor is unlikely to be of considerable importance in animal feeding. The true advantage lies in the unique combination of nutrients found in invertebrates. Given the incorporation of raw biomass into diets, its nutritional value in terms of vitamins and minerals is remarkably high. Therefore, in the commercial cultivation of invertebrates, it is essential to pay close attention to the specific composition of the nutritional substrate and its targeted enrichment.

Given the nutrient-rich nature of invertebrate biomass, we expected to observe an increase in specific nutrients (e.g., protein, vitamins, and minerals) after enriching the nutrient substrate. If the experiment was successful, it could lead to a wider use of invertebrate biomass in nutrition and animal feed, as well as to the creation of new medical products. A new method of designing the nutrient composition of invertebrate biomass by enriching the food substrate with precursors is presented as a biomass design strategy.

A common question is why is it necessary to accumulate nutrients in the biomass of invertebrates rather than simply adding them to the final food product? The nutrients present in invertebrate biomass are in a bioactive form, which increases the bioaccessibility and bioavailability of the substances in the proposed food. Insufficient bioavailability of ingested vitamins and minerals can cause significant problems in agricultural production; for example, broiler chickens face challenges in obtaining an available form of vitamin D through the direct addition of its precursor to their feed. When vitamin D is added to the diet as cholecalciferol, it must be converted by enzymatic processes in a chicken’s body to its transport form, calcidiol, and then to its active form, calcitriol. In young broilers, the enzyme system is not fully developed. Rapid growth under conditions of vitamin D deficiency, which impairs intestinal transport of calcium and phosphorus, can lead to musculoskeletal problems and increased mortality. In egg production, such a deficiency manifests itself in reduced egg laying rates, poorer egg quality, and a shortened lifespan of the hens, mainly due to increased stress on the liver and kidneys. An experiment to feed earthworms (*Eisenia*) to chickens showed not only rapid and healthy growth of the chicks, but also a significant improvement in the taste quality of the meat [50]. Similar studies have been carried out on other livestock breeds, not only in relation to vitamin D. Therefore, the accumulation of nutrients in invertebrate biomass offers a greater economic advantage in agricultural production than the direct addition of chemical precursors to livestock feed.

Enrichment of the dietary substrate with precursors does not significantly increase production costs. Vitamin and mineral supplements for livestock can serve as suitable precursors. The number of precursors required is minimal—about ten times less than for supplements added to vertebrate diets. The incorporation of the precursors into the feed substrate does not require any special operations. Typically, the feed mixture is prepared and replaced at regular intervals, and the addition of precursors to the mixture is a straightforward technological process. For certain invertebrates, such as compost worms, it is even possible to introduce the precursors in a single application when the beds are formed. The yield of enriched biomass, considering the bioactive state of the nutrients, would address animal feeding at a qualitatively different level. It is likely that such a biomass will become an alternative to the costly search for bioactive forms of specific vitamins and minerals.

## 2. Materials and Methods

### 2.1. Experimental Design

Two invertebrate species, the house cricket *Acheta domesticus* (Linnaeus, 1758) and the earthworm *Eisenia fetida* (Savigny, 1826), were chosen as model species. The cricket culture was obtained from a private entrepreneur who breeds them to supply pet shops, while the earthworms were obtained from a collection of cultures cultivated at the Department of Agricultural Biology of Tomsk University.

The experiment was conducted in five groups for each model species as follows: (1) a control where individuals were fed on a substrate with no enrichment, (2) a group developed on a substrate enriched with vitamins C and B7, (3) a group with a complex mineral supplement for plants (chelate), (4) a group with vitamins B1 (thiamin), B3 (niacin), and B9 (folate), and (5) a group based on FSV A, D, E, and K.

The cultures were raised under laboratory conditions at a temperature of c. +25 °C for both crickets and earthworms, and at a humidity of c. 60% for crickets and c. 80% in earthworms. Model species were placed in separate plastic containers and provided with a feeding substrate and precursors, or without them in the case of the control group. The feeding substrate for cockroaches contained a mixture of grated carrot (12 g), oat flakes (10 g), dried milk (1 g), and dried Gammarus (1 g), and for earthworms, it contained loose tea leaves (5 g). The containers with crickets were also provided with Petri dishes with water.

The replacement of substrate and addition of precursors were undertaken three times a week. Under a precursor (a substance inserted into the feeding substrate and shared in the metabolism of invertebrates), a particular nutrient was generated in the biomass. In the experiment the following substances were chosen as precursors: vitamins C (ascorbic acid, Cas: 50-81-7) and B7 (biotin, Cas: 58-85-5) to generate proteins; a complex mineral addition for plant (chelate) to minerals; vitamins B1 (thiamine, Cas: 59-43-8), B3 (nicotinamide, Cas: 98-92-0); and B9 (folate, Cas: 11096-55-2) for concordant vitamins of B-complex; and vitamins A (retinol, Cas: 11103-57-4), D (cholecalciferol, Cas: 67-97-0), E (d-alpha-tocopherol, Cas: 59-02-9), and K (phytomenadione, Cas: 84-80-0) for fat-soluble vitamins within approx. 95% purity.

Enrichment of the feeding substrate was generated in two stages. At the first stage, precursors were inserted in minimal doses ranging from 1 to 50 mg per 1 kg of feeding substrate according to the type of input substance. Such a dosage corresponds approximately with recommendations for vitamin and mineral rations provided to agricultural animals to prevent hypovitaminosis. At the second stage, doses of precursors were increased twice in proportion to each substance input in the substrate to attain a sufficiently enriched biomass up to the required level for metabolism, and to accumulate particular nutrients. Quantities of the input samples of the precursors are given in Table 1.

After 30 days, samples of crude frozen biomass (0.4 kg) of each model species were analyzed. At the end of the cultivation period, the earthworms were placed in water for 30 min to expel any undigested matter remaining in their gut. They were then released onto filter paper and frozen. After the feeding period, the required number of crickets were selected directly from the cultivation containers and placed in a separate container to be frozen.

The analyses were undertaken in the test centre “OOO Sibtest” as a small-scale innovative enterprise of the National Research Tomsk Polytechnic University, Tomsk, Russia in a laboratory accredited with the licence “GOSTAkkreditatsiya”, No.GOST.RU.22152. Data are presented as means and standard errors (with five replicates for each analysis). A total of 500 g of crickets and 500 g of worms were used. This equates to 2500 crickets and 1000 worms. Each group contained 50 g of invertebrates, comprising 250 crickets and 100 worms. There were five repetitions in each case, with each repetition containing 10 g of invertebrates, including 50 crickets and 20 worms. Mortality was observed in crickets during cultivation. This is a standard situation for insects with a short life cycle. These losses were offset by the large number of individuals involved in the experiment, which provided the required mass for analysis.

Sample analyses were aimed at detecting ash; carbohydrates; chitin; proteins, including content and ratio of amino acids; lipids, including analysis of fat acids; vitamins B1 (thiamine), B2 (riboflavin), B3 (niacinamide), B9 (folic acid), B12 (cyanocobalamin), A (retinol palmitate), D3 (cholecalciferol), E (α-tocopherol), and K (fillokinone); and minerals: iron (Fe), selenium (Se), zinc (Zn), magnesium (Mg), copper (Cu), manganese (Mn), phosphorus (P), lead (Pb), mercury (Hg), molybdenum (Mo), iodine (I), calcium (Ca), sodium (Na), potassium (K), and chlorine (Cl). The calorific values of the biomass for both species were also determined.

The results are presented in tabulate form according to GOST, a summary of Russian State standards, brief explanations of which, together with reference numbers in parentheses, are provided below.

***Carbohydrates*** were determined by the method of mass concentration in terms of glucose, which is based on the ability of reducing carbohydrates formed under acid hydrolysis of the sample, which reduced ferricyanide to ferrocyanide in an alkaline medium. In terms of glucose, it is determined by the titration of surplus ferricyanide in a standard glucose solution after reaction with the reducing matter.

***Proteins*** were revealed by the protein mass fraction using the Kjeldahl method, which requires the mineralisation of organic substances of a sample and subsequent determination of nitrogen as a result of the quantity of generated ammonia.

***Lipids*** were determined using a Soxhlet extractor, which involves repeatedly extracting fat from a dehydrated sample in a Soxhlet extractor. The solvent is then eliminated, and the fat is desiccated until it reaches a constant mass.

***Vitamin B1*** was determined using high-yield (high-performance) liquid chromatography, HPLC, as the total content of thiamine including its phosphorylated derivatives.

***Vitamin B2*** was also determined using high-yield (high-performance) liquid chromatography, HPLC, by riboflavin content.

***Vitamins B3 and B9*** were determined by the method of capillary electrophoresis traditionally used for water-soluble vitamins. For the detection of B3 as panteonic acid and B9 as folic acid, the method of micellar electrokinetic capillary chromatography was used. In this case, the identification and quantification of vitamins was carried out with the help of software.

***Vitamin B6*** was determined using the method of high-yield (high-performance) liquid chromatography, HPLC, as a sum of the mass fraction of pyridoxine, pyridoxal, and pyridoxamine, including their phosphorylated products in terms of pyridoxine (GOST EN 14164-2014 2017).

***Vitamin B12*** was revealed using the method of reversed-phase high-performance liquid chromatography with detection in visible spectrum under a 550 nanometre wavelength.

***Vitamin A*** was determined using the method of mass concentration of retinol, retinol acetate, and retinol palmitate under high-yield (high-performance) liquid chromatography, HPLC, with measurements ranging from 0.5 to 10.0 parts per million (ppa).

***Vitamin E*** also was determined using the HPLC method of mass concentration of α-, β-, γ- и δ- tocopherols, and figured in parts per million (ppa).

***Iron (Fe)*** was determined using the method of studying the reaction of iron ions with sulfosalicylic acid in an alkaline medium, resulting in a yellow-coloured complex compound, the intensity of which is proportional to the mass concentration of iron measured under a 400–430 nanometre wavelength ranging from 0.10 to 2.00 mg/dm^3^.

***Phosphorus (P)*** was determined using a method based on the dehumidification of the sample with further incineration, cooling, and hydrolysis of incineration residue with nitric acid, and filtration and further dilution with a mixture of ammonium monovanadate and ammonium heptamolibdate, resulting in the formation of a yellow compound, determined by photometric scaling of optical density under a 430 nanometre wavelength.

***Selenium*** (Se), ***magnesium*** (Mg), ***copper*** (Cu), ***zinc*** (Zn), and ***manganese*** (Mn) were determined by the method of inductively-coupled plasma mass spectrometry (ICP MS).

***Crude fibre*** was detected using the Genneberg and Shtoman Method, based on the serial processing of samples with acid and alkali solutions by incineration and quantitative determination of organic remains by weight. The content of crude fibre was figured as a percentage of mass concentration or as grams per 1 kg of dry matter.

***Caloricity*** was determined by calculation of the sum of caloricity of each component (carbohydrates, lipids, and proteins) of biomass per 100 g of the sample.

### 2.2. Statistical Analysis

R version 4.0.2 [55] was used for statistical analysis of the nutrient parameters. For analysis of changes in nutrient composition in the dependence of the feeding substrate, Student’s *t*-test for independent samples was applied. Evaluation of significance of differences with control and confidence intervals was estimated with correction of multiple comparisons according to the Bonferroni method of correction; in case of 5 comparisons, α = 0.01, and in case of 8 comparisons, α = 0.006. Accordingly, confidence intervals (CI) in the graphs are given with 99% and 99.4% correction for probability.

## 3. Results

### Nutrient Composition of the Biomass of Model Species

The nutrient composition of the two model species differs significantly. Comparative analysis of B vitamin content in the control group of crickets and earthworms revealed low indices of B1, B2, and B3 and high levels of B9 and B12 in crickets, whereas earthworms showed high indices of B1, B2, and B3 and low levels of B9 and B12 (Figure 1, Appendix A). Vitamin B6 levels were comparable in both species.

A single dose of substrate enrichment with biotin (B7) and vitamin C, minerals, and fat-soluble and B-complex vitamins did not alter the levels of vitamins B1, B2, and B3, but a statistically significant increase in vitamins B6, B9, and B12 was observed compared to the control group (Appendix A). As a result of enriching the substrate with a double dose of B vitamins, the content of B6 increased twofold and B12 increased threefold (Appendix A).

A statistically significant increase in B2, B3, B6 and B12 vitamins and a decrease in B9 vitamins is registered in earthworms after enrichment of the substrate with a single dose of biotin (B7) and C vitamins, minerals, and fat-soluble and B complex vitamins, and B1 content is increased only after enrichment of the substrate with microelements (*p* < 0.01) (Appendix A). The increase in all vitamins, except B3 and B1 with levels comparable to the control group was registered after substrate enrichment with a double dose of B vitamins (*p* < 0.01) (Appendix A).

The general trends of nutrient composition changes after substrate enrichment in crickets and earthworms are comparable and characterized by an increase in specific B vitamins from an initial high level.

The comparative analysis of the content of fat-soluble vitamins revealed a high index of vitamins A and D3 in crickets and a low index in earthworms, and a higher content of vitamins E and K in earthworms in the control (Appendix A). Among the FSV, there was an increase in vitamin A and stable, non-variable levels of vitamins D3, E, and K after single-dose substrate fortification with biotin (B7) and vitamin C, minerals, FSV, and B-complex vitamins. The double-dose fortification resulted in an increase in all FSV compared with the control group (Figure 2). Overall, the double dose of fat-soluble vitamin enrichment increased all vitamins in the group in both crickets and earthworms (*p* < 0.01).

The level of vitamin C in the control groups of model species is 1.5 times lower in crickets. After enrichment of the substrate, the vitamin C content of crickets does not change, whereas it increases significantly (*p* < 0.01) in earthworms (Figure 3).

Comparative analysis of macro-elements showed high levels of Cl and P in crickets, but the levels of other elements in crickets and earthworms were similar to the control group. A single-dose addition of biotin (B7) and vitamin C, minerals, and fat-soluble and B-complex vitamins to the dietary substrate had no effect on the nutrient composition of crickets but significantly reduced the level of Cl in earthworms. Double-dose enrichments decreased the levels of K and P in both crickets and earthworms but significantly increased the level of Cl in earthworms (Figure 4, Appendix A).

Overall, substrate enrichments did not result in significant increases in macro-elements in crickets and earthworms, but double-dose enrichment may decrease the level of macro-elements in the biomass of the model species.

The comparative analysis of the microelement content in the control group shows significant differences between crickets and earthworms; in crickets, Cu, I, Mn, Mo, and Zn are higher, while in earthworms, Fe, Hg, Pb, and Se are higher (Figure 5, Appendix A). After a single dose of substrate enrichment, the content of Cu, Fe, I, Mn, Pb, and Se did not change significantly. As a result of substrate enrichment with B-vitamins, FSV, and minerals, the Hg content increased significantly, and the Mo content decreased (*p* < 0.01) (Appendix A). The double dose of mineral enrichment resulted in a significant decrease in the number of microelements except Fe and Se, which were similar to the control (*p* < 0.01).

In earthworms, a single dose of substrate enrichment significantly decreased the level of Pb and Se, but indexes of other elements remained comparable to the control. The double dose of enrichment has reduced the levels of Hg, I, Pb, and Se, and other elements are comparable with the control (*p* < 0.01).

Thus, mineral enrichment had no effect on the microelement composition of crickets, whereas Pb and Se were significantly reduced in earthworms. The double-dose enrichment showed a tendency to decrease the mineral content in the biomass of the model species.

In the control groups of crickets and earthworms, the protein content is similar, and the water content is double that in earthworms and the other parameters; namely, ash, fats, carbohydrates, cellulose, and chitin are significantly higher in crickets (Figure 6). No differences in the content of ash, protein, carbohydrates, cellulose, and water are registered after single- and double-dose enrichment of the feeding substrate of model species of invertebrates.

A single dose of substrate enrichment has no effect on fat content, but a double dose significantly increases the fat content of crickets by more than double. After a single dose of biotin (B7) and FSV, chitin content increases by a factor of two, but with a double dose, the chitin content is comparable to the control.

The lipid content of earthworms increases only after single and double doses of mineral supplementation. The content of several nutrients in earthworms decreases with a single dose and increases with a double dose. Overall, the content of major organic nutrients is higher in crickets than in earthworms in both the control and experimental groups. The protein content is comparable in both model species.

The enrichment of the feeding substrate has no significant effect on the change in the main parameters of the nutrient composition in the invertebrate biomass. Only a double dose of enrichment leads to a significant increase in fat in crickets and in chitin in earthworms.

The calorific value of crickets is twice that of earthworms and substrate enrichment has no significant effect on the calorific value of the model species (Figure 7).

## 4. Discussion

Both crickets and earthworms have traditionally been used as research subjects in the field of agriculture [56,57,58,59,60,61,62,63]. Since the house cricket *Acheta domesticus* Linnaeus, 1758 has been authorized by the European Commission (EC) for sale, farming, and consumption as novel food [37,38,39], many articles on the nutrient composition of crickets are published nowadays. A comparative analysis of the nutrient composition in the biomass of different cricket species used as food in the world is given in [45]. It is stated that only *Acheta domesticus* accumulates a higher level of protein in its biomass (2.41–71.09 g/100 g dry weight), and *Gryllus assimilis* and *G. bimaculatus* are able to accumulate up to 70 g/100 g dry weight, although a typical level of protein in crickets is about 50 g/100 g dry weight. *Acheta testacea* is a species similar to the house cricket that accumulates the lowest level of protein, 18 g/100 g dry weight. Thus, the nutrient indices of different taxonomically similar species within a group of Orthoptera differ greatly. Nevertheless, different cricket species are considered as potential sources of dietary protein, minerals, and vitamins [45,64].

Earthworms are also used as food in some regions of Southeast Asia [46], but mainly, the nutrient composition of the biomass of *Eisenia* sp. has been studied to justify the effectiveness of the use of earthworms in the production of feed for livestock (broilers, fish, pigs, etc.) [50,51,52]. The protein content of the worm powder obtained from *E. fotida* was found to be six times higher than that of barley meal (66.90% compared to 11.81%). Feeding broilers with a mixture of traditional feed and worm powder improves meat flavour and increases weight gain [51]. In guinea pigs, the energy exchange and digestibility of the diet is improved by 10% [52]. In fish and crustaceans, it provides essential fatty acids [51].

Traditionally, data on the nutrient composition of invertebrates are considered to be constant as a result of being obtained after a particular survey of species biomass. However, there may be differences within the nutrient composition or within its level, caused by several reasons; for example, significant differences in the nutrient composition of the earthworm *Eisenia foetida* have been reported, even when different methods of biomass fixation are used [65]. Oven-dried biomass can preserve a lower level of protein and some minerals but a high level of fatty acids, whereas freeze-dried biomass has shown a higher level of protein and can demonstrate a better detection of nutrients in the same biomass. Interspecific differences in biomass content were also found in related groups of invertebrates. Comparison of the nutrient composition of the house cricket *Acheta domesticus* and the field cricket *Gryllus bimaculatus* showed a twofold higher content of water, ash, and carbohydrates, but a twofold lower content of fats and fibres in the house cricket [64]. Obviously, the differences in nutrient composition are typical for invertebrate species belonging to different taxonomic ranks or occupying different ecological niches. As a result of the experiment, the differences in composition and level of nutrient content in the biomass of two model species are considered logical, but the exact nutrient parameters are considered more interesting.

Contrasting levels of B vitamins in crickets and earthworms—with low levels of vitamins B1, B2, and B3 in crickets and high levels in earthworms, and vice versa, with high levels of B9 and B12 in crickets and low levels in earthworms—may be explained by internal physiological reasons of these species, since the enrichment of the substrate did not modify the vitamin levels, which were initially low. It is worth noting that enrichment by at twice the dosage increased the level of vitamin B, which was high in the control. It should be noted that it is possible to increase the level of vitamins B in crickets and earthworms, but each species is supplied with a high level of specific vitamins, typical of each species.

The FSV showed a different picture in terms of content after substrate enrichment. The minimal dose increased the levels of vitamins A and D3, which were initially present at high levels in crickets and low levels in earthworms; so, even this minimal enrichment of the substrate can increase their levels in the model species. The double-dose enrichment showed a strong increase in the fat-soluble vitamins in the biomass of crickets and earthworms, which represents a good perspective for the enrichment of the invertebrate biomass by the fat-soluble vitamins.

Vitamin C levels were higher in the control group of earthworms and increased after substrate enrichment, i.e., only earthworms are able to accumulate vitamin C in significant quantities, but in crickets, vitamin C levels remained stable throughout the experiments. This can probably be explained by the fact that earthworms grow throughout their lives and can constantly increase in size, which requires a high protein content in the biomass. Vitamin C is involved in the process of protein synthesis and is essential for earthworms. Crickets in both the larval and imago stages are covered with a cuticular sheath that does not allow them to increase biomass immediately without ecdysis, so the level of vitamin C is stable and sufficient to provide a vital function of an organism, except for biomass expansion.

In contrast to vitamins, mineral levels remain stable in the model species. Neither a single nor a double dose of precursors has any effect on increasing the mineral content, but a double dose can even decrease the mineral content of the biomass. This fact confirms that the accumulation of minerals after a change in feeding substrate is not typical for invertebrates, whose organisms try to maintain “mineral homeostasis”, which defines a low level of heavy metals and some elements that could cause a toxic effect. In this case, the biomass of invertebrates can be considered more valuable and healthier than that of fish, for example, since some fish species are able to accumulate heavy metals and toxins, resulting in an organism being poisoned [66,67,68,69].

The differences in the content of the main nutrients in crickets and earthworms can be seen in the high content of ash, fats, carbohydrates, cellulose, and chitin in crickets and the twice-higher content of water in earthworms, which can be explained by the morphophysiology of these species. Thus, earthworms do not have a hard and massive chitin cover and need more water to ensure the elasticity of the body and the realization of biochemical reactions, while crickets are provided with a high level of chitin in an external skeleton to maintain the homeostasis of an organism with a low water content.

From a nutritional point of view, it is argued that earthworm biomass is more nutritious, given the similar protein content in the model species, because its protein is not surrounded by hard and indigestible chitin. It is likely that the morphophysiological organization of invertebrates with hard external coverings does not allow such a significant increase in the main substance parameters without changing their morphological indices, such as body growth and cuticle thinning. Apparently, this fact also determines the stable calorific value indices of both species. The biomass of crickets is twice as calorific as that of earthworms, and the enrichment of the food substrate does not affect its increase or decrease.

Contrasting indices of B-vitamins, FSV, vitamin C, and microelement content were recorded in both the control and experimental groups of crickets and earthworms. In crickets and earthworms, similar trends of increasing B-vitamin levels from initially high levels in the biomass could be observed after enrichment of the diet with B-vitamin precursors.

Double-dose enrichment in the diet resulted in an increase in all fat-soluble vitamins in the biomass of crickets and earthworms.

In the control group, the content of vitamin C was 1.5 times lower in crickets than in earthworms. After the enrichment of the food substrate, the level of vitamin C in crickets was not changed, while in earthworms, this level significantly increased. The levels of all the macroelements studied did not change significantly after enrichment of the diet, but a double dose of enrichment resulted in a decrease in the levels of macroelements in both crickets and earthworms.

The content of microelements did not change after single-dose enrichment, but some microelements, Pb and Se, were significantly reduced in earthworms after double-dose enrichment of the feeding substrate. Double-dose enrichment of the feed substrate reduced the mineral content of the invertebrate biomass.

The organic content is higher in crickets in both the control and experimental groups, but the protein content is comparable. Substrate enrichment has no effect on the organic fraction of nutrients in the biomass of the model species, except that double-dose enrichment increases the fat content in crickets and the chitin content in earthworms. The calorific value of crickets is twice that of earthworms and does not change significantly after double-dose enrichment.

It is shown that the content of vitamins in the biomass of the model species can be controlled within the process of substrate enrichment. Enrichment with a high dose of nutrient precursors does not always result in an increase in their biomass level, but sometimes the opposite effect is observed.

Key nutrient changes between the control and enriched groups are shown in Table 2.

The enrichment of the food substrate with precursors of specific nutrients showed an uneven accumulation of nutritional elements in the biomass of model invertebrate species. It was practically impossible to increase the protein content, except in worms, where its level slightly increased, which can be explained by the morphophysiology of worms having soft cuticles that increase with the growth of the individual. Enrichment of the food substrate with minerals also did not show a clear pattern of dependence of accumulation of individual minerals in biomass on the introduction of precursors into the food substrate. This could be because worms are more efficient at accumulating minerals and the enrichment of the substrate may slightly increase the content of some elements, but to a negligible extent.

The high concentration of vitamins in crickets can be attributed to their active lifestyle and short life cycle, which means that individuals of all ages are present in the biomass. In contrast, the size and age of worms are similar, and they lead a slow lifestyle.

Nevertheless, both worms and insects can significantly alter nutrient composition. This can be applied in practice; for example, an intensive supply of vitamins and minerals in bioactive form is required when raising young animals. Additionally, the invertebrate protein currently available in powder form for baking is somewhat depleted in vitamins and minerals. Developing methods to preserve the active forms of these substances in biomass could significantly enrich animal feed and diets.

## 5. Conclusions

Enriching the diet with a double dose of nutrient precursors significantly increases the levels of fat-soluble vitamins and some B-group vitamins in crickets and earthworms.

Vitamin C enrichment led to a significant increase in vitamin C levels in earthworms, whereas crickets showed no change.

Adding a single dose of minerals to the food substrate did not significantly change their content, but double-enrichment of the food substrate reduced the mineral content of the invertebrate biomass.

Single-substrate enrichment did not affect the organic fraction of nutrients in the biomass of the model species, whereas double enrichment increased fat content in crickets and chitin content in earthworms.

It is shown that the content of vitamins in the biomass of model species can be controlled in the process of substrate enrichment. Enrichment with a high dose does not always lead to an increase in the level of their biomass; sometimes, the opposite effect is observed.

The experiment only studied two species of invertebrates in laboratory conditions, which imposes certain limitations. When implementing this practice in farming, it is possible that adjustments to substrate enrichment will be required. It is also necessary to study how other species used in industry react to substrate enrichment. The way they accumulate nutrients may differ. Future research should focus on identifying the bioactive forms of nutrients in invertebrate biomass and testing the effects of enriched biomass on young livestock and poultry. Scaling up the experiment is also necessary to identify species-specific features of the accumulation of specific nutrients in invertebrates. Selecting forms capable of maximizing the accumulation of necessary nutrients is likely to contribute to the development of new invertebrate breeds and this sector of animal husbandry.

This is the seventh publication in Tomsk State University’s project “Invertebrates of Siberia as a promising source of animal protein for innovative feed and food production”. The aim of the project is to identify and recommend for cultivation invertebrates that can be cultivated at minimal cost and produce a substantial yield under Siberian conditions [15,16,17,49,53,54].

## Figures and Tables

**Figure 1 insects-16-00632-f001:**
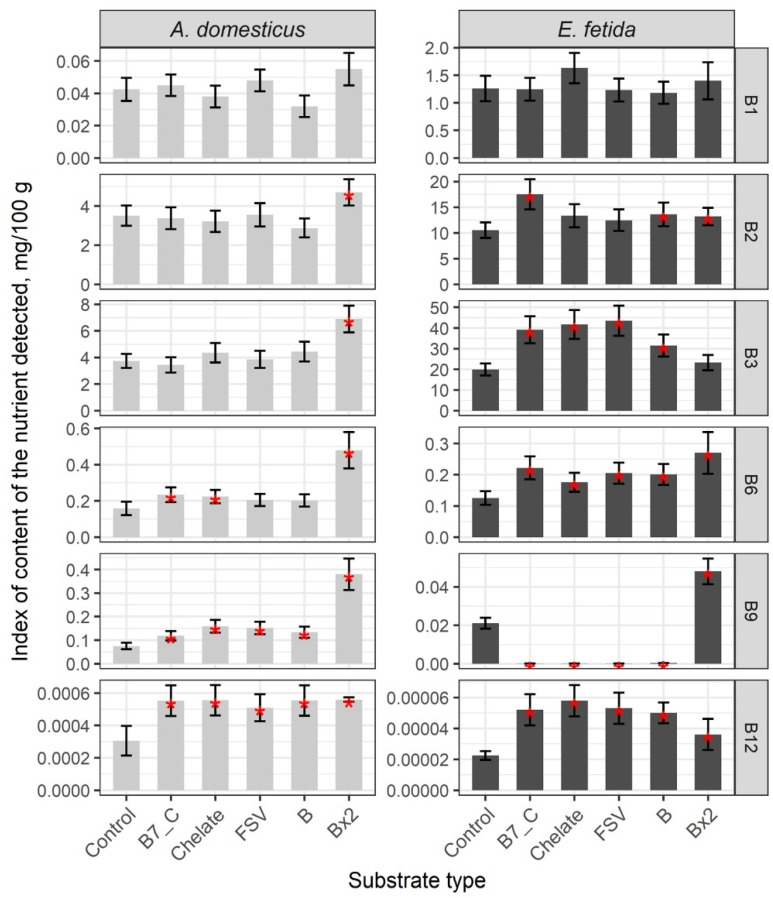
Content of B vitamins in biomass of the house cricket *Acheta domesticus* (Linnaeus, 1758) and the earthworm *Eisenia fetida* (Savigny, 1826) fed with different substrates. Average indexes are given on graphs, with confidence intervals (with correction for multiple comparisons CI 99%). Designations: Control—control group; B7_C—substrate enriched with a single dose of vitamins B7 and C; Chelate—substrate enriched with a single dose of mineral complex additive; FSV—substrate enriched with a single dose of FSV; B—substrate enriched with a single dose of B-complex vitamins (B1, B3, and B9); Bx2—substrate enriched with a double dose of B-complex vitamins (B1, B3, and B9). Red star indicates statistically significant differences with Control, *p* < 0.01.

**Figure 2 insects-16-00632-f002:**
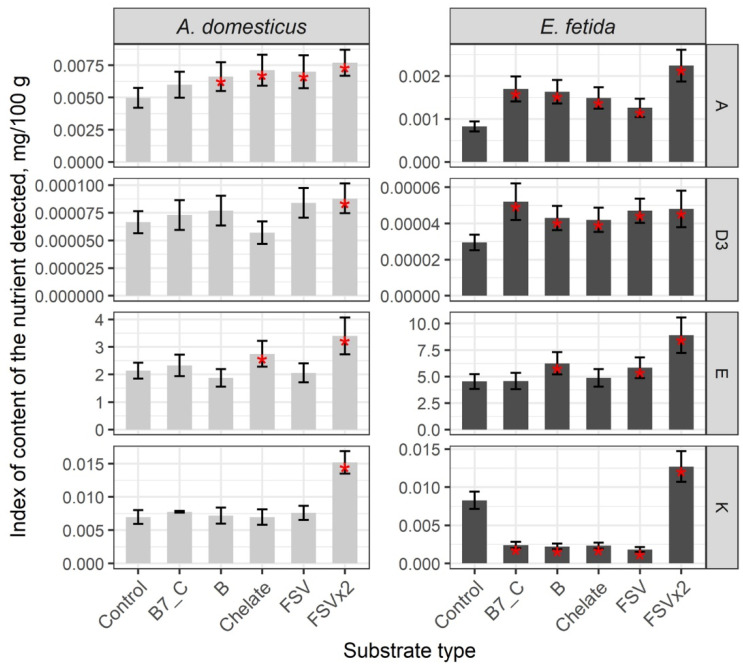
Content of FSV in biomass of the house cricket *Acheta domesticus* (Linnaeus, 1758) and the earthworm *Eisenia fetida* (Savigny, 1826) fed on different substrates. Average indexes are given on graphs, with confidence intervals (with correction for multiple comparisons CI 99%). Designations: Control—control group; B7_C—substrate enriched with a single dose of vitamins B7 and C; Chelate—substrate enriched with a single dose of mineral complex additive; FSV—substrate enriched with a single dose of FSV; FSVx2—substrate enriched with double dose of FSV; B—substrate enriched with a single dose of B-complex vitamins (B1, B3 and B9). Red star indicates statistically significant differences with Control, *p* < 0.01.

**Figure 3 insects-16-00632-f003:**
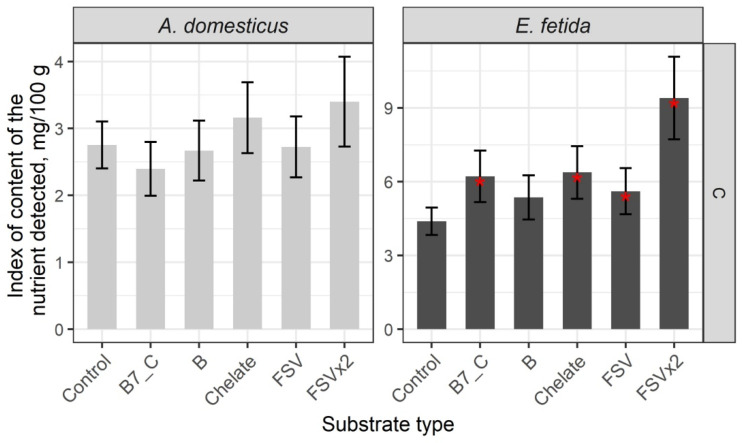
Content of C vitamin in biomass of the house cricket *Acheta domesticus* (Linnaeus, 1758) and the earthworm *Eisenia fetida* (Savigny, 1826) fed on different substrates. Average indexes are given on graphs, with confidence intervals (with correction for multiple comparisons CI 99%). Designations: Control—control group; B7_C—substrate enriched with a single dose of vitamins B7 and C; Chelate—substrate enriched with a single dose of mineral complex additive; FSV—substrate enriched with a single dose of FSV; FSVx2—substrate enriched with double dose of FSV; B—substrate enriched with a single dose of B-complex vitamins (B1, B3, and B9). Red star indicates statistically significant differences with Control, *p* < 0.01.

**Figure 4 insects-16-00632-f004:**
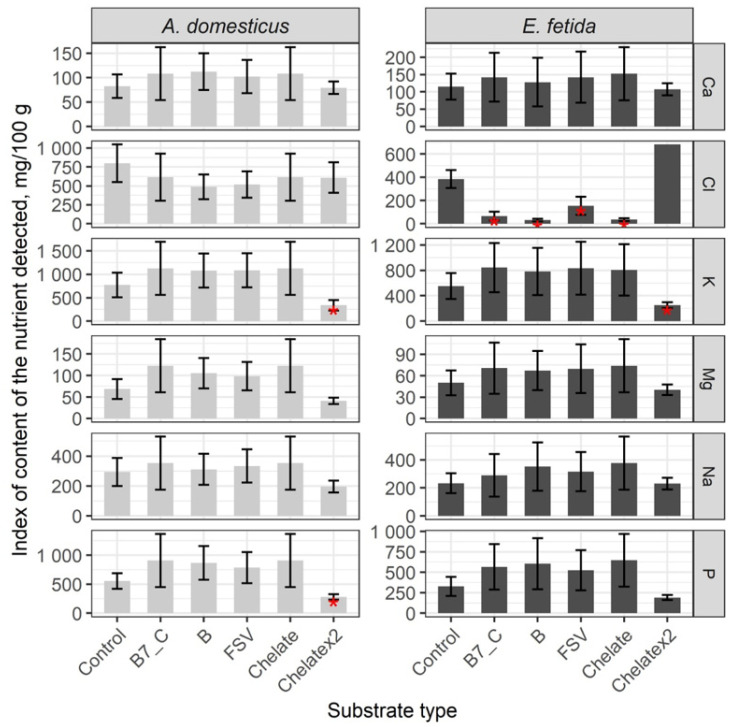
Content of macroelements in biomass of the House cricket *Acheta domesticus* (Linnaeus, 1758) and the earthworm *Eisenia fetida* (Savigny, 1826) fed on different substrates. Average indexes are given on graphs, with confidence intervals (with correction for multiple comparisons CI 99%). Designations: Control—control group; B7_C—substrate enriched with a single dose of vitamins B7 and C; Chelate—substrate enriched with a single dose of mineral complex additive; Chelatex2—substrate enriched with double dose of mineral complex additive; FSV—substrate enriched with a single dose of FSV; B—substrate enriched with a single dose of B-complex vitamins (B1, B3, and B9). Red star indicates statistically significant differences with Control, *p* < 0.01.

**Figure 5 insects-16-00632-f005:**
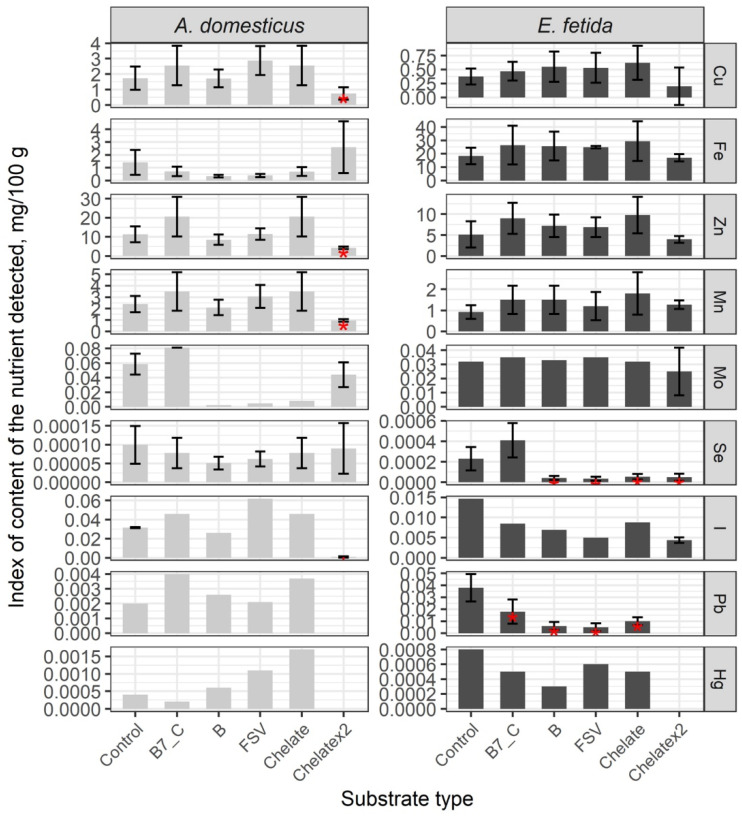
Microelements content in the House cricket *Acheta domesticus* (Linnaeus, 1758) and the earthworm *Eisenia fetida* (Savigny, 1826) fed on different substrates. Average indexes are given on graphs, with confidence intervals (with correction for multiple comparisons CI 99%). Designations: Control—control group; B7_C—substrate enriched with a single dose of vitamins B7 and C; Chelate—substrate enriched with a single dose of mineral complex additive; Chelatex2—substrate enriched with double dose of mineral complex additive; FSV—substrate enriched with a single dose of FSV; B—substrate enriched with a single dose of B-complex vitamins (B1, B3, and B9). Red star indicates statistically significant differences with Control, *p* < 0.01.

**Figure 6 insects-16-00632-f006:**
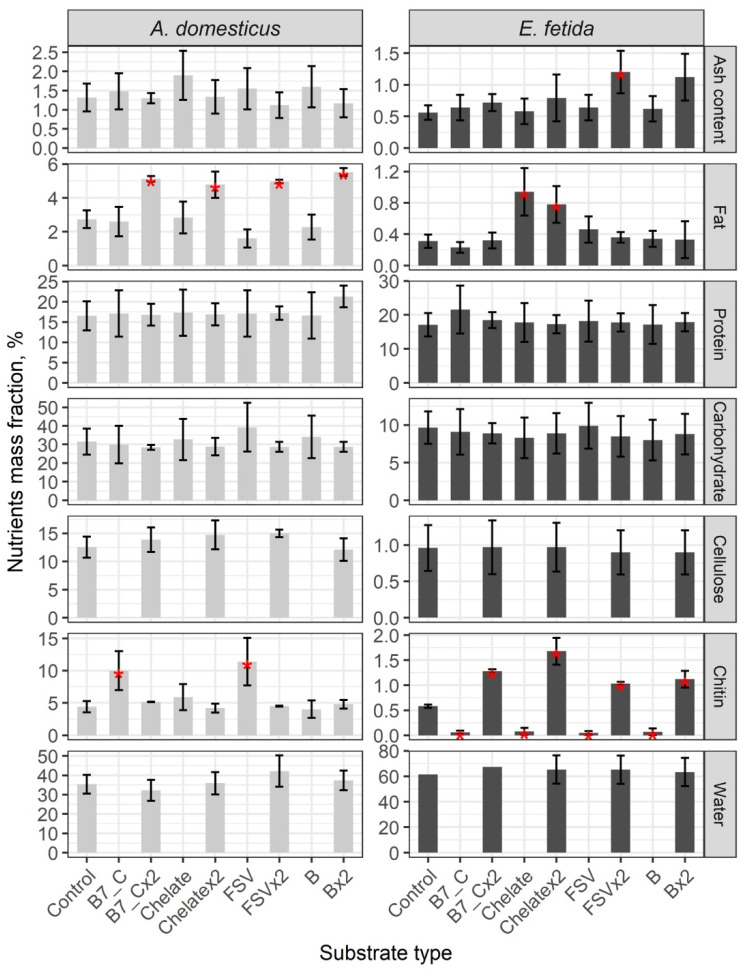
General complex analysis of biomass of the house cricket *Acheta domesticus* (Linnaeus, 1758) and the earthworm *Eisenia fetida* (Savigny, 1826) fed on different substrates. Average indexes are given on graphs, with confidence intervals (with correction for multiple comparisons CI 99.4%). Designations: Control—control group; B7_C—substrate enriched with a single dose of vitamins B7 and C, B7x2_C—substrate enriched with doubled dose of vitamins B7 and C; Chelate—substrate enriched with a single dose of mineral complex additive; Chelatex2—substrate enriched with doubled dose of mineral complex additive; FSV—substrate enriched with a single dose of FSV; FSVx2—substrate enriched with double dose of FSV; B—substrate enriched with a single dose of B-complex vitamins (B1, B3, and B9); Bx2—substrate enriched with double dose of B-complex vitamins (B1, B3, and B9). Red star indicates statistically significant differences with Control, *p* < 0.01.

**Figure 7 insects-16-00632-f007:**
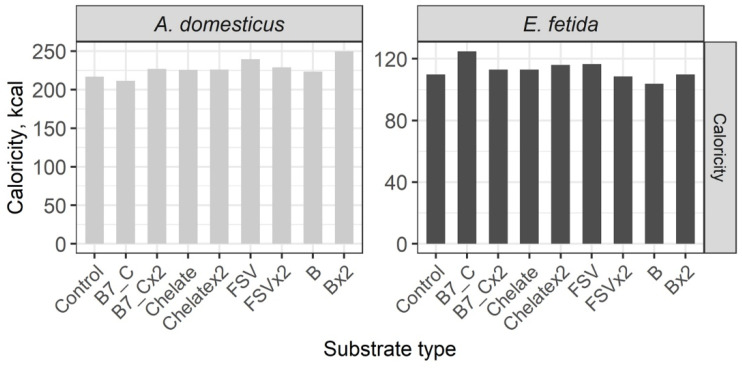
Calorific value of the House cricket *Acheta domesticus* (Linnaeus, 1758) and the earthworm *Eisenia fetida* (Savigny, 1826) fed on different substrates. Average indexes are given on graphs. Designations: Control—control group; B7_C—substrate enriched with a single dose of vitamins B7 and C; B7x2_C—substrate enriched with doubled dose of vitamins B7 and C; Chelate—substrate enriched with a single dose of mineral complex additive; Chelatex2—substrate enriched with double dose of mineral complex additive; FSV—substrate enriched with a single dose of FSV; FSVx2—substrate enriched with double dose of FSV; B—substrate enriched with a single dose of B-complex vitamins (B1, B3, and B9); Bx2—substrate enriched with double dose of B-complex vitamins (B1, B3, and B9).

**Table 1 insects-16-00632-t001:** Quantity of precursors added to feeding substrate of model species during the experiment.

Type of Precursor	I Stage, Single Dose of Precursor			II Stage, Double Dose of Precursor		
	Per 1 kg of Food Substrate, mg	Per Food Portion of Crickets, mg	Per Food Portion of earthworms, mg	Per 1 kg of Food Substrate, mg	Per Food Portion of Crickets, mg	Per Food Portion of Earthworms, mg
C	50	1.2	0.25	100	2.4	0.5
B7	25	0.6	0.125	50	1.2	0.25
Minerals (chelate)	5	0.12	0.025	10	0.24	0.05
B1	2	0.048	0.01	4	0.096	0.02
B3	30	0.72	0.15	60	1.44	0.3
B9	1	0.024	0.005	1	0.048	0.01
A	25	0.6	0.125	50	1.2	0.25
D	2.5	0.06	0.0125	5	0.12	0.025
E	20	0.48	0.1	40	0.96	0.2
K	2	0.048	0.01	4	0.096	0.02

**Table 2 insects-16-00632-t002:** Nutrients whose levels increased significantly with changes in feeding substrate.

Substrate	B12	B2	B3	B6	B9	A	D3	E	K	C
	*A. domesticus*									
**Control**	**0.00031**	**3.51**	**3.75**	**0.16**	**0.08**	**0.0050**	**0.000067**	**2.14**	**0.0070**	**2.75**
B7_C	0.00055	–	–	0.23	0.12	–	–	–	–	–
Chelate	0.00056	–	–	0.22	0.16	0.0071	–	2.75	–	–
B	0.00055	–	–	–	0.13	0.0066	–	–	–	–
Bx2	0.00056	4.70	6.90	0.48	0.38	–	–	–	–	–
FSV	0.00051	–	–	–	0.15	0.0070	–	–	–	–
FSVx2	–	–	–	–	–	0.0077	0.000088	3.40	0.0152	–
** *E. fetida* **										
**Control**	**0.00002**	**10.53**	**19.92**	**0.13**	**0.021**	**0.0008**	**0.000030**	**4.54**	**0.0083**	**4.39**
B7_C	0.00005	17.52	39.14	0.22	–	0.0017	0.000052	–	–	6.21
Chelate	0.00006	–	41.72	0.18	–	0.0015	0.000042	–	–	6.38
B	0.00005	13.61	31.53	0.20	–	0.0016	0.000043	6.25	–	–
Bx2	0.00004	13.20	–	0.27	0.048	–	–	–	–	–
FSV	0.00005	–	43.54	0.21	–	0.0013	0.000047	5.83	–	5.61
FSVx2	–	–	–	–	–	0.0022	0.000048	8.90	0.0127	9.40

Note: – no significant differences with control.

## Data Availability

The original contributions presented in this study are included in the article. Further inquiries can be directed to the corresponding authors.

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
