# Peer review of "Invertebrates of Siberia, a Potential Source of Animal Protein for Innovative Food and Feed Production: Biomass Nutrient Composition Change in the Earthworm Eisenia fetida (Savigny, 1826) and the House Cricket Acheta domesticus (Linnaeus, 1758)"

_insects, 2025, doi:10.3390/insects16060632_

Round 1
Reviewer 1 Report (Previous Reviewer 2)
Comments and Suggestions for Authors
This manuscript investigates a method to modulate the nutrient composition of biomass in the house cricket (Acheta domesticus) and the earthworm (Eisenia fetida) through feed substrate enrichment. The study holds significant scientific and practical value in the context of invertebrates as sustainable protein sources, particularly for food and feed industries. However, to further enhance the quality of the manuscript, I recommend revisions in the following areas:
- Materials and Methods Section
- On page 5, line 191, it is stated that each analysis was performed with five replicates, but the specific setup for biological replicates is not clarified. Providing details on the number of animals per group and the replication design would enhance the transparency of the experimental setup.
- On page 4, lines 159–160, temperature and humidity conditions are mentioned, but other environmental factors, such as light cycles, are not specified. Including these details would help assess the impact of experimental conditions on the results.
- On page 5, line 187, it is noted that 0.4 kg of frozen biomass was used for analysis, but the specific sample processing procedures are not described. Adding details on sample collection and storage would ensure consistency in sample handling.
- Results Section
- Providing the complete dataset as supplementary material would enhance the transparency and reproducibility of the study.
- Figure 7 shows no significant changes in calorific value. To avoid redundancy, consider integrating this data into Table 2 or supplementary material.
- Discussion Section
- Incorporating a discussion on the mechanisms underlying nutrient accumulation would strengthen the depth of the analysis and provide greater insight into the observed results.
This study provides valuable data on nutrient enrichment in invertebrates but requires revisions to improve methodological details, data completeness, discussion depth, and language clarity.
Author Response
Comments and Suggestions for Authors
This manuscript investigates a method to modulate the nutrient composition of biomass in the house cricket (Acheta domesticus) and the earthworm (Eisenia fetida) through feed substrate enrichment. The study holds significant scientific and practical value in the context of invertebrates as sustainable protein sources, particularly for food and feed industries. However, to further enhance the quality of the manuscript, I recommend revisions in the following areas:
- Materials and Methods Section
- On page 5, line 191, it is stated that each analysis was performed with five replicates, but the specific setup for biological replicates is not clarified. Providing details on the number of animals per group and the replication design would enhance the transparency of the experimental setup.
A total of 500 grams of crickets and 500 grams of worms were used. This equates to 2,500 crickets and 1,000 worms.
Each group contained 50 grams of invertebrates, comprising 250 crickets and 100 worms. There were five repetitions in each case, with each repetition containing 10 grams of invertebrates, including 50 crickets and 20 worms. Mortality was observed in crickets during cultivation. This is a standard situation for insects with a short life-cycle. These losses were offset by the large number of individuals involved in the experiment, which provided the required mass for analysis.
- On page 4, lines 159–160, temperature and humidity conditions are mentioned, but other environmental factors, such as light cycles, are not specified. Including these details would help assess the impact of experimental conditions on the results.
Other environmental factors were not considered, as the experiment was conducted under standardised laboratory conditions for all groups studied. Light cycles were not important for the worms, which were grown in containers in the dark. Light is also not important for crickets, as they are crepuscular. The cricket containers were kept in an unlit place.
- On page 5, line 187, it is noted that 0.4 kg of frozen biomass was used for analysis, but the specific sample processing procedures are not described. Adding details on sample collection and storage would ensure consistency in sample handling.
At the end of the cultivation period, the earthworms were placed in water for 30 minutes to expel any undigested matter remaining in their guts. They were then released onto filter paper and frozen. After the feeding period, the required number of crickets were selected directly from the cultivation containers and placed in a separate container to be frozen.
- Results Section
- Providing the complete dataset as supplementary material would enhance the transparency and reproducibility of the study.
Tables containing all the results are available to download separately from the ‘Supplementary Material’.
- Figure 7 shows no significant changes in calorific value. To avoid redundancy, consider integrating this data into Table 2 or supplementary material.
We cannot include this data in Table 2, as this table only contains data showing statistically significant differences. However, we do not consider these graphs to be superfluous, since statistically insignificant differences can also be of scientific and practical interest.
- Discussion Section
- Incorporating a discussion on the mechanisms underlying nutrient accumulation would strengthen the depth of the analysis and provide greater insight into the observed results.
Thank you very much for the recommendation. We will work on interpreting this material in our next project.
This study provides valuable data on nutrient enrichment in invertebrates but requires revisions to improve methodological details, data completeness, discussion depth, and language clarity.
Submission Date
06 May 2025
Date of this review
24 May 2025 06:56:22
The present version of the text has been linguistically revised and corrected by Professor Mark Seaward of the University of Bradford, United Kingdom.
We are extremely grateful for your careful attention to our work. The questions you asked have helped us not only to improve the manuscript, but also to plan our future work. Unfortunately, we were unable to provide a complete revision of the text for a few questions, primarily due to conflicts with the requirements of other reviewers. However, we hope that this fifth version of the manuscript, incorporating over ten rounds of recommendations from reviewers, will be eligible for publication in the journal. With deep gratitude,
The Authors
Reviewer 2 Report (New Reviewer)
Comments and Suggestions for Authors
The manuscript explores a timely and relevant topic within the journal’s scope: the potential to modulate the nutritional composition of invertebrate biomass through targeted enrichment of the feeding substrate. The combination of Acheta domesticus and Eisenia fetida as model organisms is appropriate, and the analytical methodology employed—including GOST standards and accredited protocols—is technically sound.
Nevertheless, the manuscript would benefit from several improvements. First, although this is part of an established research line from the same group, the specific novelty of this study should be more explicitly articulated, especially in relation to previous works already published on Nauphoeta cinerea and Lissachatina fulica. Readers unfamiliar with earlier papers should be able to understand what is new here.
The quality of the English can be improved throughout, particularly in the Introduction and Discussion sections, where the text becomes repetitive or imprecise. A professional language review is recommended to enhance clarity and readability.
Methodologically, it would be helpful to include more detail about the number of individuals per replicate and whether any physiological effects (e.g., mortality, behavior, growth) were observed during the enrichment period. The rationale for the specific precursor doses should also be justified more clearly, particularly in relation to known nutritional requirements or metabolic thresholds.
The statistical approach is acceptable, but complementing the t-test comparisons with a factorial ANOVA could provide a clearer view of species × treatment interactions. Figures are clear, but legends could be expanded to better explain all abbreviations, and Table 2 would be more informative if it included data on variables that showed no significant change.
The discussion could be more focused on the biological relevance and implications of the findings. For instance, the increase in fat or chitin content following double-dose enrichment is interesting but requires contextualization in terms of practical impact on feed quality or digestibility.
Author Response
Comments and Suggestions for Authors
The manuscript explores a timely and relevant topic within the journal’s scope: the potential to modulate the nutritional composition of invertebrate biomass through targeted enrichment of the feeding substrate. The combination of Acheta domesticus and Eisenia fetida as model organisms is appropriate, and the analytical methodology employed—including GOST standards and accredited protocols—is technically sound.
Nevertheless, the manuscript would benefit from several improvements. First, although this is part of an established research line from the same group, the specific novelty of this study should be more explicitly articulated, especially in relation to previous works already published on Nauphoeta cinerea and Lissachatina fulica. Readers unfamiliar with earlier papers should be able to understand what is new here.
This study is novel in that it experimentally verifies the accumulation capacity of invertebrates of different taxonomic types. While a previous study (Tshernyshev et al., 2024a) investigated insects and molluscs, this study investigated insects and worms. Interestingly, under equal maintenance conditions, the accumulation of specific nutrients differs between insects, worms, and molluscs. Specifically, insects accumulate vitamins more effectively, worms accumulate minerals more effectively, and molluscs accumulate ash more effectively.
The quality of the English can be improved throughout, particularly in the Introduction and Discussion sections, where the text becomes repetitive or imprecise. A professional language review is recommended to enhance clarity and readability.
The present version of the text has been linguistically revised and corrected by Professor Mark Seaward of the University of Bradford, United Kingdom.
Methodologically, it would be helpful to include more detail about the number of individuals per replicate and whether any physiological effects (e.g., mortality, behavior, growth) were observed during the enrichment period. The rationale for the specific precursor doses should also be justified more clearly, particularly in relation to known nutritional requirements or metabolic thresholds.
A total of 500 grams of crickets and 500 grams of worms were used. This equates to 2,500 crickets and 1,000 worms.
Each group contained 50 grams of invertebrates, comprising 250 crickets and 100 worms. There were five repetitions in each case, with each repetition containing 10 grams of invertebrates, including 50 crickets and 20 worms. Mortality was observed in crickets during cultivation. This is a common occurrence for insects with a short life-cycle. These losses were compensated for by the large number of individuals involved in the experiment, which provided the required mass for analysis.
The statistical approach is acceptable, but complementing the t-test comparisons with a factorial ANOVA could provide a clearer view of species × treatment interactions. Figures are clear, but legends could be expanded to better explain all abbreviations, and Table 2 would be more informative if it included data on variables that showed no significant change.
We do not consider it appropriate to overload this study with unnecessary statistical calculations. The legends are detailed, and the captions under the figures explain everything. Table 2 summarises the results and clearly indicates which differences were significant.
The discussion could be more focused on the biological relevance and implications of the findings. For instance, the increase in fat or chitin content following double-dose enrichment is interesting but requires contextualization in terms of practical impact on feed quality or digestibility.
Thank you very much for the recommendation. We will work on interpreting this material in our next project.
Submission Date
06 May 2025
Date of this review
16 May 2025 21:31:21
We are extremely grateful for your careful attention to our work. The questions you asked have helped us not only to improve the manuscript, but also to plan our future work. Unfortunately, we were unable to provide a complete revision of the text for a few questions, primarily due to conflicts with the requirements of other reviewers. However, we hope that this fifth version of the manuscript, incorporating over ten rounds of recommendations from reviewers, will be eligible for publication in the journal. With deep gratitude,
The Authors
Reviewer 3 Report (New Reviewer)
Comments and Suggestions for Authors
Dear Authors, please find my detailed feedback aimed at refining and strengthening your manuscript for its next revision.
Title and Abstract
- The title is informative but overly long. Consider shortening it while retaining clarity, e.g., "Biomass Nutrient Modulation in Eisenia fetida and Acheta domesticus via Substrate Enrichment."
- The abstract reiterates points from the main text. It should be more concise, focusing on the key objectives, methods, and core results with less repetition.
- The novelty claim “new method” lacks specificity. Clarify what aspect is truly novel—biomass design strategy, targeted vitamin enrichment, or species-specific accumulation patterns?
- While the abstract mentions increases in nutrient levels, it lacks key quantitative metrics that would help readers quickly assess the scale of change.
- The abstract says that double-dose enrichment "may have the opposite effect of reducing some nutrients," but it does not specify which nutrients.
Introduction
- While the background is comprehensive, it is excessively long. Shorten this by removing generic statements and secondary context (e.g., paragraphs on edible snails and livestock processing innovations).
- The rationale for using invertebrates is repeated multiple times in various ways (food crisis, nutrient density, environmental footprint). A more concise argument with references would be sufficient.
- The research gap and specific research questions are not clearly outlined. Explicitly state what prior studies lacked and how this study fills those gaps.
- While the paper uses two model species, the reasoning for choosing Acheta domesticus and Eisenia fetida over others like Gryllus bimaculatus or Tenebrio molitor should be strengthened, ideally with citations supporting their relevance or contrasting nutrient profiles.
- No hypotheses or expected directional outcomes are stated. Adding hypotheses for vitamin and mineral accumulation would enhance scientific rigor.
Materials and Methods
- The experiment involves 5 treatment groups with two dosage stages, but this structure is not clearly visualized or tabulated. A treatment matrix table would be helpful.
- The cricket and earthworm substrates differ greatly (carrot, oats, milk vs. tea leaves), which introduces confounding effects. How was comparability ensured across such contrasting substrates?
- The justification for precursor dosage scaling (double the recommended value for livestock) lacks references or physiological logic tied to invertebrate metabolism.
- It is unclear whether vitamin losses during substrate preparation or invertebrate digestion were accounted for.
- Although GOST methods are mentioned, there is no justification for why these particular assays (e.g., acid hydrolysis for carbohydrates, ICP-MS for metals) are suitable or how their accuracy compares with international standards like AOAC or ISO.
- The Bonferroni correction is stated, but there is no clarity on how repeated measures or intergroup variance was handled. Were the five replicates biological or technical? ANOVA might be more suitable than multiple t-tests.
Results
- The figures are detailed, but the text only restates visual content. Instead, integrate interpretation and highlight biologically meaningful patterns in the narrative.
- Some results are described in raw numbers (e.g., “vitamin B6 increased twofold”) while others use vague terms like “significant increase.” Provide all values with % change and p-values.
- Phrases like “content increased significantly” should specify whether increases are statistically and nutritionally meaningful.
- For many nutrients, the starting baseline values are omitted, making it hard to understand the magnitude of change post-enrichment.
- The conclusion that “substrate enrichment did not significantly affect nutrient composition” is misleading, as certain vitamins (e.g., B12, A, D) did show notable increases.
- Figures lack units on axes (e.g., μg/g dry weight), error bars in some cases, and are overly cluttered, making comparative reading difficult.
Discussion
- The discussion lacks mechanistic insight into why certain nutrients accumulated more in crickets vs. earthworms. Consider discussing species-specific metabolic pathways or gut microbiota influences.
- Some nutrients decreased (e.g., Pb, Se, K) with double enrichment. Are these desirable reductions or unintended depletions? This should be interpreted biologically.
- Statements like “this biomass will become an alternative to the costly search for bioactive forms” are overreaching without economic analysis or industry comparisons.
- Although some references are used for protein content comparisons, broader global data on insect-based enrichment strategies would strengthen the context.
- The discussion misses out on translating findings into feed formulation practices, policy recommendations, or commercial scalability.
Conclusion
- Much of the conclusion restates results without synthesizing broader implications or identifying the main takeaway.
- This section should outline the limitations—e.g., short experimental duration, species scope, or environmental variability.
- There is a need to propose future research directions—e.g., scaling experiments, testing with livestock feeding trials, or exploring gut absorption of bioactive compounds.
The manuscript contains numerous grammatical errors, awkward phrasing, and overlong sentences that hinder readability. A professional language edit is advised.
Author Response
Comments and Suggestions for Authors
Dear Authors, please find my detailed feedback aimed at refining and strengthening your manuscript for its next revision.
Title and Abstract
- The title is informative but overly long. Consider shortening it while retaining clarity, e.g., "Biomass Nutrient Modulation in Eisenia fetida and Acheta domesticus via Substrate Enrichment."
Thank you for recommending that we change the title of the work. The paper presented is the seventh in a series of studies planned under the university-supported project. In order to produce a successful report, however, we need to keep the title consistent with those of the previous six studies. We sincerely hope that the reviewer will agree to this request, as other reviewers have done, and allow us to keep the title unchanged.
- The abstract reiterates points from the main text. It should be more concise, focusing on the key objectives, methods, and core results with less repetition.
Sorry, this is the fifth version of the text, reworked in accordance with the reviewers' recommendations, including those regarding the 'Abstract'. According to the Journal's rules, the 'Abstract' should be comprehensive and explain all aspects of the work. The 'Short Summary' is intended to reflect the main purpose of the work, as you recommend for the 'Abstract'. This is presented in the current version of the manuscript.
- The novelty claim “new method” lacks specificity. Clarify what aspect is truly novel—biomass design strategy, targeted vitamin enrichment, or species-specific accumulation patterns?
The phrase is defined as follows:
'A new method of designing the nutrient composition of invertebrate biomass by enriching the food substrate with precursors is presented as a biomass design strategy.'
- While the abstract mentions increases in nutrient levels, it lacks key quantitative metrics that would help readers quickly assess the scale of change.
According to the other reviewers' recommendation, this data was eliminated from the abstract, with reference to its presence in the text.
- The abstract says that double-dose enrichment "may have the opposite effect of reducing some nutrients," but it does not specify which nutrients.
This is shown in lines 33-35 in the phrase “ The content of a wide range of minerals did not change after single dose enrichment, but some microelements such as Pb and Se decreased significantly in earthworms after double‐dose enrichment of the feed substrate”.
Introduction
- While the background is comprehensive, it is excessively long. Shorten this by removing generic statements and secondary context (e.g., paragraphs on edible snails and livestock processing innovations).
Thank you for your recommendations to revise and shorten the introduction. It has already been considerably shortened, but additional paragraphs were added at the request of previous reviewers. Unfortunately, some of your recommendations conflict with those of other reviewers, so we have had to revise the 'Introduction' and 'Results' sections to take their opinions into account, for which we apologise.
- The rationale for using invertebrates is repeated multiple times in various ways (food crisis, nutrient density, environmental footprint). A more concise argument with references would be sufficient.
Currently, there is a lot of discussion in the mass media in favour of using alternative protein (for example, https://youtu.be/DLTzH1FcePA). We have attempted to highlight that the use of invertebrates meets the current demands of modern life. Readers can choose for themselves ‘A more concise argument.’
- The research gap and specific research questions are not clearly outlined. Explicitly state what prior studies lacked and how this study fills those gaps.
Previous studies have mainly focused on the protein and other nutrient content of invertebrates that are traditionally farmed or can be farmed at minimal cost. However, no studies have been conducted on the targeted modification of individual nutrients in biomass. The proposed study demonstrates that, while it has some limitations, targeted modification of nutrient composition is possible. This opens up the prospect of creating feeds, food products and medicinal forms with the highest possible concentration of required substances.
- While the paper uses two model species, the reasoning for choosing Acheta domesticusand Eisenia fetida over others like Gryllus bimaculatus or Tenebrio molitor should be strengthened, ideally with citations supporting their relevance or contrasting nutrient profiles.
Many terrestrial invertebrates are tested for their suitability as food and feed. Our study involves experimentally testing the accumulation capacity of invertebrates of various taxonomic types. While a previous study (Tshernyshev et al., 2024a) investigated insects and molluscs, this study investigated insects and worms. Interestingly, under equal maintenance conditions, the accumulation trends of specific nutrients differ between insects, worms, and molluscs. Specifically, insects accumulate vitamins more effectively, worms accumulate minerals more effectively, and molluscs accumulate ash more effectively. Insects are twice as calorific as snails and worms, and the calorific content of insects increases with the addition of nutrients, while that of snails and worms remains unchanged. Our experiments have shown that the content of vitamins, fats, proteins and carbohydrates in invertebrate biomass, as well as the caloric content of insects, can be controlled by enriching the feeding substrate with precursors. No special attachment to other species, such as Gryllus bimaculatus or Tenebrio molitor, was required.
- No hypotheses or expected directional outcomes are stated. Adding hypotheses for vitamin and mineral accumulation would enhance scientific rigor.
Given the nutrient-rich nature of invertebrate biomass, we expected to observe an increase in specific nutrients (e.g. protein, vitamins and minerals) after enriching the nutrient substrate. If the experiment is successful, it could lead to wider use of invertebrate biomass in nutrition and animal feed, as well as in the creation of new medical products.
Materials and Methods
- The experiment involves 5 treatment groups with two dosage stages, but this structure is not clearly visualized or tabulated. A treatment matrix table would be helpful.
Tables containing all the results are available to download separately from the ‘Supplementary Material’.
- The cricket and earthworm substrates differ greatly (carrot, oats, milk vs. tea leaves), which introduces confounding effects. How was comparability ensured across such contrasting substrates?
While the substrates are very different from one another, they are standardised for each species of invertebrate. In this case, the absorption of additional nutrients into the body is comparable between crickets and worms and to the control.
- The justification for precursor dosage scaling (double the recommended value for livestock) lacks references or physiological logic tied to invertebrate metabolism.
Yes, this is an experimental study that could be investigated further if necessary.
- It is unclear whether vitamin losses during substrate preparation or invertebrate digestion were accounted for.
This study only recorded the final accumulation of vitamins and minerals in the biomass of invertebrates. Losses during digestion or in the substrate were not considered.
- Although GOST methods are mentioned, there is no justification for why these particular assays (e.g., acid hydrolysis for carbohydrates, ICP-MS for metals) are suitable or how their accuracy compares with international standards like AOAC or ISO.
GOST methods are provided to explain the analytical techniques employed. These methods are standardised with classical chemical research. No comparison with international standards such as AOAC or ISO was required or carried out.
- The Bonferroni correction is stated, but there is no clarity on how repeated measures or intergroup variance was handled. Were the five replicates biological or technical? ANOVA might be more suitable than multiple t-tests.
The t-test was used to compare the control groups with those of the invertebrate culture supplemented with different food additives. The substrate was the same in both control and experimental groups, the only difference being the addition of nutrients in the experimental group. The significance level for multiple comparisons was adjusted using the Bonferroni method. Statistical analysis between species was not performed. The use of ANOVA was not possible in this case as the data were presented as means and standard errors (with five replicates for each analysis). We do not consider it appropriate to overload this study with unnecessary statistical calculations. The legends are detailed and everything is explained in the captions under the figures. Table 2 summarises the results obtained and clearly shows which differences were significant.
Results
- The figures are detailed, but the text only restates visual content. Instead, integrate interpretation and highlight biologically meaningful patterns in the narrative.
corrected
- Some results are described in raw numbers (e.g., “vitamin B6 increased twofold”) while others use vague terms like “significant increase.” Provide all values with % change and p-values.
corrected
- Phrases like “content increased significantly” should specify whether increases are statistically and nutritionally meaningful.
corrected
- For many nutrients, the starting baseline values are omitted, making it hard to understand the magnitude of change post-enrichment.
The magnitude of change post-enrichment was defined as differential between results in each group and control group meaning
- The conclusion that “substrate enrichment did not significantly affect nutrient composition” is misleading, as certain vitamins (e.g., B12, A, D) did show notable increases.
Corrected:
“substrate enrichment did not significantly affect on some nutrient composition”
- Figures lack units on axes (e.g., μg/g dry weight), error bars in some cases, and are overly cluttered, making comparative reading difficult.
All the axes are designated and refer to two species of invertebrates.
Discussion
- The discussion lacks mechanistic insight into why certain nutrients accumulated more in crickets vs. earthworms. Consider discussing species-specific metabolic pathways or gut microbiota influences.
- Some nutrients decreased (e.g., Pb, Se, K) with double enrichment. Are these desirable reductions or unintended depletions? This should be interpreted biologically.
- Statements like “this biomass will become an alternative to the costly search for bioactive forms” are overreaching without economic analysis or industry comparisons.
- Although some references are used for protein content comparisons, broader global data on insect-based enrichment strategies would strengthen the context.
- The discussion misses out on translating findings into feed formulation practices, policy recommendations, or commercial scalability.
A new paragraph is added:
The enrichment of the food substrate with precursors of specifi c nutrients showed an uneven accumulation of nutritional elements in the biomass of model invertebrate species. It was practically impossible toincrease the protein content, except in worms, where its level slightly increased, which can be explained by the morphophysiology of worms having soft cuticles, which can increase with the growth of the individual. Enrichment of the food substrate with minerals also did not show a clear pattern of dependence of accumulation of individual minerals in biomass on the introduction of precursors into the food substrate. Perhaps, worms are more effi cient at accumulating minerals, and enrichment of the substrate may slightly increase the content of some elements, but to a negligible extent.
The high concentration of vitamins in crickets can be attributed to their active lifestyle and short life cycle, which means that individuals of all ages are present in the biomass. In contrast, the size and age of worms are similar, and they lead a slow lifestyle.
Nevertheless, both worms and insects can significantly alter nutrient composition. This can be applied in practice. For example, an intensive supply of vitamins and minerals in bioactive form is required when raising young animals. Additionally, the invertebrate protein currently available in powder form for baking is somewhat depleted in vitamins and minerals. Developing methods to preserve the active forms of these substances in biomass could significantly enrich animal feed and diets.
Conclusion
- Much of the conclusion restates results without synthesizing broader implications or identifying the main takeaway.
- This section should outline the limitations—e.g., short experimental duration, species scope, or environmental variability.
- There is a need to propose future research directions—e.g., scaling experiments, testing with livestock feeding trials, or exploring gut absorption of bioactive compounds.
A new paragraph is added:
The experiment only studied two species of invertebrates in laboratory conditions, which imposes certain limitations. When implementing this practice in farming, it is possible that adjustments to substrate enrichment will be required. It is also necessary to study how other species used in industry react to substrate enrichment. The way they accumulate nutrients may differ. Future research could focus on identifying the bioactive forms of nutrients in invertebrate biomass, and testing the effects of enriched biomass on young livestock and poultry. Scaling up the experiment is also necessary in order to identify species-specific features of the accumulation of specific nutrients in invertebrates. Selecting forms capable of maximising the accumulation of necessary nutrients is likely to contribute to the development of new invertebrate breeds and this sector of animal husbandry.
Comments on the Quality of English Language
The manuscript contains numerous grammatical errors, awkward phrasing, and overlong sentences that hinder readability. A professional language edit is advised.
Submission Date
06 May 2025
Date of this review
22 May 2025 18:05:46
The present version of the text has been linguistically revised and corrected by Professor Mark Seaward of the University of Bradford, United Kingdom.
We are extremely grateful for your careful attention to our work. The questions you asked have helped us not only to improve the manuscript, but also to plan our future work. Unfortunately, we were unable to provide a complete revision of the text for a few questions, primarily due to conflicts with the requirements of other reviewers. However, we hope that this fifth version of the manuscript, incorporating over ten rounds of recommendations from reviewers, will be eligible for publication in the journal. With deep gratitude,
The Authors
Round 2
Reviewer 3 Report (New Reviewer)
Comments and Suggestions for Authors
I am satisfied with the revisions the authors made and recommend to accept the manuscript in its current form.
This manuscript is a resubmission of an earlier submission. The following is a list of the peer review reports and author responses from that submission.
Round 1
Reviewer 1 Report
Comments and Suggestions for Authors
Dear Authors,
Your revised manuscript entitled “Nutritional value of banded cricket and mealworm larvae” has been sent for a third review.
Thanks for reading my comments and updating the manuscript. There are still a few things left open and I would like to mention them briefly
- You noted that your statistical evaluation was confined to using a t-test. Scientifically speaking, this is problematic without full access to all primary data, and careful wording is required when discussing significance in the text. E.g. line 301: You write that there is a significant difference between the two model species, but no evidence from significance tests is provided -just as an example
- Line 162-164: The wording is linguistically unusual. Minerals in the strict sense are not precursors for molecules. I wish to point out again that manufacturing mineral supplements for animal feed or food involves an intricate process, and additives are subject to legal approval. As such, I believe the statement ‘They do not require special preparation’ does not reflect reality.
- Figure 6: For cellulose and water, it looks as if some bars are missing from the diagram – or are the values just very small?
- Line 526: In Figure 5, the red asterisks next to specific minerals/trace elements seem to indicate statistical differences relative to CTR cells. Does this require any adjustment to the statement you made?
- I acknowledge your intention to end the project with a successful seventh publication; however, I must stress that while the manuscript delves into feeding-related enhancements of model species and their macro/micronutrient profiles, it does not address how these findings meet nutrient intake requirements for animals or humans. This gap is unfortunate given the availability of EFSA data. I will pass this critique on to the editorial board for their consideration.
xxx
Author Response
Dear Reviewer!
We are grateful for the time you have taken to review our work. The submitted version represents a general revision of the manuscript in accordance with the reviewers' requirements. We have made corrections wherever possible. We hope to answer the remaining questions in future studies.
Yours sincerely, authors
Reviewer 2 Report
Comments and Suggestions for Authors
This manuscript investigates the feasibility of modulating the nutrient composition of biomass from house crickets (Acheta domesticus) and earthworms (Eisenia fetida) through enriched feeding substrates, aiming to provide a sustainable animal protein source for innovative food and feed production. The topic holds scientific merit and practical potential, particularly in the context of rising global protein demands. The authors demonstrate the effects of substrate enrichment on vitamins, minerals, and caloric value, supported by extensive data. However, the manuscript requires improvement in structure, data presentation, and language to enhance readability and academic rigor. Specific Recommendations:
- Introduction: The introduction is overly lengthy and includes tangential details unrelated to the core topic. It should be concise (1-1.5 pages), focusing on the potential of insect and earthworm biomass as protein sources and the need for nutrient modulation, with the research objective stated upfront.
- Separation of Results and Discussion: The “Results” section contains interpretive content (e.g., analysis of vitamin C differences) that belongs in the “Discussion.” The “Results” should be restricted to data description, with all interpretations moved to a dedicated “Discussion” section for logical clarity.
- Data Presentation: Figures (Figs. 1-6) are data-dense but lack a summary overview. Consider adding a table in the “Results” or “Discussion” section to list key nutrient changes between control and enriched groups, facilitating quick comprehension of findings.
- Conclusions: The current conclusions repeat results. Simplify to 3-4 key findings (e.g., “Double-dose enrichment significantly increases fat-soluble vitamin content”) and include actionable suggestions (e.g., “Double-dose enrichment is recommended to optimize vitamin levels in feed”).
- Additional Improvements:
-Reformat the manuscript per the journal’s template; currently, sections lack numbering and the layout appears disorganized.
-Define abbreviations upon first use in the text (e.g., “FSV = fat-soluble vitamins”).
-Capitalization in the title should be adjusted (e.g., “of” should not be capitalized, revise to “Invertebrates of Siberia…”).
This study offers novelty but needs structural refinement, enhanced data presentation, and language improvement.
Author Response
Dear Reviewer!
We are grateful for the time you have taken to review our work. The submitted version represents a general revision of the manuscript in accordance with the reviewers' requirements. We have made corrections wherever possible. Unfortunately, some of your recommendations are contrary to those of other reviewers, so we have had to revise the 'Introduction' and 'Results' to take account of their opinions, for which we apologise.
We sincerely hope that we have been able to take your recommendations into account and correct the manuscript to make it more informative.
Yours sincerely, the authors